# WEAKLY-SUPERVISED DOMAIN ADAPTATION IN FEDERATED LEARNING FOR HEALTHCARE

## ABSTRACT

Federated domain adaptation (FDA) describes the setting where a set of source clients seek to optimize the performance of a target client. To be effective, FDA must address some of the distributional challenges of Federated learning (FL). For instance, FL systems exhibit distribution shifts across clients. Further, labeled data are not always available among the clients. To this end, we propose and compare novel approaches for FDA, combining the few labeled target samples with the source data when auxiliary labels are available to the clients. The in-distribution auxiliary information is included during local training to boost out-of-domain accuracy. Also, during fine-tuning, we devise a simple yet efficient gradient projection method to detect the valuable components from each source client model towards the target direction. The extensive experiments on healthcare datasets show that our proposed framework outperforms the state-of-the-art unsupervised FDA methods with limited additional time and space complexity.

## 1 INTRODUCTION

Federated learning (FL) is a distributed learning paradigm, where an aggregated model is learned using local decentralized data on edge devices (McMahan et al., 2017). FL systems usually share the model weights or gradient updates of clients to the server, which prevents direct exposure of the sensitive client data. As a result, data heterogeneity remains an important challenge in FL, and much of the research focuses on mitigating the negative impacts of the distribution shifts between clients' data (Wang et al., 2019; Karimireddy et al., 2020; Xie et al., 2020b). Further, much of the FL literature has focused on settings where all datasets are fully-labeled. However, in the real world, one often encounters settings where the labels are scarce on some of the clients. To this end, multi-source domain adaptation (MSDA) (Ben-David et al., 2010; Zhao et al., 2020; Guan & Liu, 2021) is a common solution to this problem, where models trained on several labeled, separate source domains are transferred to the unlabeled or sparsely labeled target domain. Here, we consider the more specialized setting of Federated domain adaptation (FDA) – where a set of source clients seek to optimize the performance of a target client. As an analogue to MSDA, one may consider clients data as different domains. Thus, goal is to learn a good model for the few-labeled target client data samples by transfer the useful knowledge from multiple source clients. In this work, we consider the FDA problem under weak supervision, where auxiliary labels are available to the clients. In brief, we propose novel approaches to deal with weakly-supervised FDA, focusing on techniques that adapt both the *local training* and *fine-tuning* stages.

**Motivating Application.** Our work is inspired and applied to applications in predictive modeling for healthcare where there can be significant differences across hospitals, causing transfer errors across sites (Li et al., 2020; Guan et al., 2021; Wolleb et al., 2022). Unlike many other industries, healthcare in the US is highly heterogeneous (e.g., HCA, the largest consortium of hospitals covers $< 2\%$ of the market (Statista, 2020; Wikipedia contributors, 2022)), thus many variables are not standardized (Adnan et al., 2020; Osarogiagbon et al., 2021). Hence, we consider experiments that simulate differences across institutions as a large shift. Further, we consider an FL application across several hospitals located at different states in the US. In this setting, FDA is employed to improve performance at a target hospital by leveraging information from all of the source hospitals. The human cost of labeling the images is expensive, thus the data are sparsely labeled. Further, in addition to the medical images, the data also include demographic information such as age, sex, race, among others. While such auxiliary data is ubiquitous, it is often ignored when working to

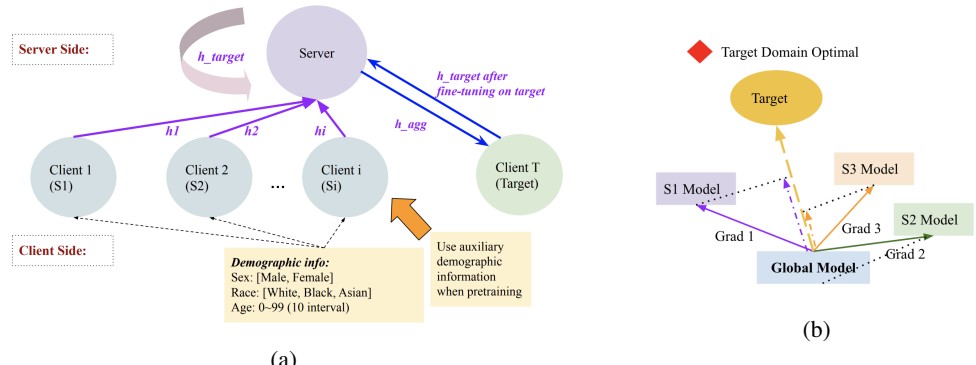

Figure 1: (a) **Proposed framework of weakly-supervised FDA**: we set up a MTL framework leveraging auxiliary labels during the source clients' local training. (b) **Intuition of GP**: we project the valuable components of source gradients towards the target direction to boost FDA performance.

improve centralized or federated models. Here, we show how this data can be used to significantly improve out-of-domain (OOD) performance when *properly* utilized in FL. Also, in FL, local models are gradually trained, so the importance of source domains may *change dynamically* during each iteration. Our work seeks to extract the valuable components of *model updates* instead of the whole models towards the target direction during each round. We discover that the few labeled target samples can provide sufficient guidance towards adapting quickly to the target domain.

Our contributions are twofold, in local training and fine-tuning. First, we leverage auxiliary information to reduce the task risk on the target client during local training. Auxiliary information is often cheap and easy to find along with the image inputs, which may offer some useful signals for the unlabeled samples, because of the underlying correlations between auxiliary tasks and main task. Xie et al. (2021) and Wang et al. (2022) both found that in centralized setting, using auxiliary information can improve OOD performance, resulting in a smaller OOD risk compared with the baseline. In our work, we set up a *cheap and efficient* multi-task learning (MLT) framework with auxiliary tasks during local training of source clients, as shown in Figure 1a. By optimizing the main and auxiliary task losses together, we show empirically that one can boost target domain accuracy after fine-tuning on labeled target samples (Section 5.2). The gains are more evident when the distribution shifts are large, yet auxiliary tasks may introduce unexpected noise when domains are too close to each other.

Secondly, we observe that including auxiliary information alone does not fully account for the importance of source domains. Thus, during fine-tuning, we propose a simple yet efficient *gradient projection* (GP) method. This method utilizes the useful source domain components and projects them towards the target direction. As shown in Figure 1b, during each communication round, we compute the model updates of source clients, and then project them on the target model update obtained by fine-tuning on a small set of labeled target samples. In this way, we greedily approach to the optimum of target domain: the positive cosine similarity between a pair of (source, target) updates serves as the importance of that domain. Our experiment results indicate that this gradient projection method achieves a better and more stable FDA performance on the target client, through combining projected gradients with the fine-tuning gradient. We show the superiority of the gradient projection method through the comprehensive experiments on both medical and general-purposed datasets in Section 5.2 and Appendix A.6.

Combining two techniques together, our proposed framework outperforms the state-of-the-arts unsupervised FDA methods with limited additional computational cost. Also, we show empirically that our framework is resistant to data imbalances on the real-world MIDRC dataset (Section 5.2).

## 2 PROBLEM SETUP

In this section, we introduce the framework of weakly-supervised FDA. We first provide the background knowledge of weakly-supervised MSDA. Following this, we extend the framework to federated learning setting.

**Weakly-supervised MSDA.** Let $\mathcal{D}_S$ and $\mathcal{D}_T$ denote source and target domains. In each domain, we have sample pairs of $x \in \mathbb{R}^d$ as the input, $y \in \mathbb{R}$ as the output label. In MSDA, we have $N$ source domains $\mathcal{D}_S = \{\mathcal{D}_{S_i}\}_{i=1}^N$ with $n_i$ labeled samples as well as a target domain with $n_T$ total samples, which consists of $n_l$ labeled samples and $n_T - n_l$ (a large number of) unlabeled samples. During pretraining, we train a model $h_{S_i}$ on each source domain using the corresponding labeled samples. The goal of fine-tuning is to learn a target model $h_T$, which minimizes the target risk $\epsilon_T(h) := \mathrm{Pr}_{(x,y)\sim\mathcal{D}_T}[h(x) \neq y]$. A common approach is to first aggregates the model parameters using $\sum_{i=1}^N \alpha_i h_{S_i}$ with $\alpha_i$ controlling the importance of each source domain such that $\sum_{i=1}^N \alpha_i = 1$. Then, we fine-tune the aggregated parameters using the set of labeled samples from $\mathcal{D}_T$.

**Federated problem setting.** We now extend weakly-supervised MSDA to the FL setting. As shown in Figure 1a and Algorithm 1, we assume there are $N + 1$ clients in the system, where $N$ clients $\{\mathcal{C}_{S_i}\}_{i=1}^N$ have labeled data $\{\mathcal{D}_{S_i}\}_{i=1}^N$ and the remaining client $\mathcal{C}_T$ has $\mathcal{D}_T$ with $n_l$ labeled data. Different from the centralized setting, weakly-supervised FDA requires the target client $\mathcal{C}_T$ to have no direct access to the source data $\{\mathcal{D}_{S_i}\}_{i=1}^N$. The aggregation of the source domain models is performed by the server $\mathcal{S}$ using any federated aggregation rules such as FedAvg (McMahan et al., 2017) or SCAFFOLD (Karimireddy et al., 2020). After $\mathcal{S}$ finishes the aggregation, it sends the model parameters to $\mathcal{C}_T$ and performs a fine-tuning – just as with weakly-supervised MSDA. The final model is sent to all the source clients $\{\mathcal{C}_{S_i}\}_{i=1}^N$, which ends one communication round.

---

**Algorithm 1** Weakly-Supervised Federated Domain Adaptation

---

**Input**: $N$ source domains $\mathcal{D}_S = \{\mathcal{D}_{S_i}\}_{i=1}^N$; target domain $\mathcal{D}_T$; $N$ source clients $\{\mathcal{C}_{S_i}\}_{i=1}^N$; target client $\mathcal{C}_T$; target model $h_T^{(r-1)}$ from the previous round $r - 1$; aggregation rule $aggr$; server $\mathcal{S}$.

**Output**: Target model $h_T^{(r)}$ at round $r$.

1: **Step 1**: Local training on $\{\mathcal{C}_{S_i}\}_{i=1}^N$
2: **for** $\mathcal{D}_{S_i}$ in $\mathcal{D}_S$ **do**
3:     Initialize $h_{S_i}^{(r)} \leftarrow h_T^{(r-1)}$.
4:     Optimize $h_{S_i}^{(r)}$ on $\mathcal{D}_{S_i}$ with classification task loss (Eq. 1)
5:     Send $h_{S_i}^{(r)}$ to $\mathcal{S}$
6: **end for**
7: **Step 2**: $\mathcal{S}$ received all $\{h_{S_i}^{(r)}\}_{i=1}^N$
8: $h_{global}^{(r)} \leftarrow aggr(\{h_{S_i}^{(r)}\}_{i=1}^N)$
9: Send $h_{global}^{(r)}$ to $\mathcal{C}_T$
10: **Step 3**: $\mathcal{C}_T$ received $h_{global}^{(r)}$
11: Optimize $h_{global}^{(r)}$ on $\mathcal{D}_T$ using labeled samples.
12: $h_T^{(r)} \leftarrow h_{global}^{(r)}$
13: Send $h_T^{(r)}$ to $\mathcal{S}$
14: **Step 4**: $\mathcal{S}$ received $h_T^{(r)}$
15: **for** $\mathcal{C}_{S_i}$ in $\mathcal{C}_S$ **do**
16:     Send $h_T^{(r)}$ to $\mathcal{C}_{S_i}$
17: **end for**

---

## 3    Leveraging Auxiliary Information during Local Training

After defining the framework for weakly-supervised FDA, we now look into the ways to boost the target client performance. In this section, we explain the idea of *leveraging auxiliary information*: optimizing the main and auxiliary tasks together during source clients' local training. Here, we only derive the loss objective for one source client $\mathcal{C}_{S_i}$ (since it is the same for all source clients). Let $n$ be the total local training sample size, and $\ell$ as the loss function for the main task. Eq. 1 is the loss objective of the main task (image classification).

$$L_{main} = \sum_{j=1}^{n} \ell(h(x_j), y_j) \tag{1}$$

Let $K$ be the number of auxiliary tasks, and $z_k \in \mathbb{R}^T$, $b_k$ be $k$th auxiliary input and output labels, respectively. Thus, $z_{kj}$ and $b_{kj}$ denote the $j$th sample input and output for $k$th auxiliary task. Each auxiliary task shares the same parameters with $h$ except for the last layer (denoted as $h^{l-1}$) and we define $g_k$ to be the feature mapping from the feature representation to the $k$th auxiliary output. Lastly, let $\ell_{aux}$ as the loss function for auxiliary tasks. Then, we can construct the loss objective for auxiliary tasks as follows:

$$L_{aux} = \sum_{j=1}^{n} \sum_{k=1}^{K} \ell_{aux}(g_k(h^{l-1}(z_{kj})), b_{kj}) \tag{2}$$

In the end, we write the total loss objective for leveraging auxiliary information as Eq. 3, where $\alpha$ controls the weight of auxiliary task losses:

$$L_{aux-info} = L_{main} + \alpha \cdot L_{aux} = \sum_{j=1}^{n} \left( \ell(h(x_j), y_j) + \alpha \cdot \sum_{k=1}^{K} \ell_{aux}(g_k(h^{l-1}(z_{kj})), b_{kj}) \right) \tag{3}$$

Including auxiliary information only changes the local training process (**Step 1**): $\mathcal{C}_{S_i}$ optimizes its model parameters using a summed loss together with auxiliary task losses. As shown in Figure 1a, we optimize the losses of all tasks together, which consist of negative/positive diagnosis as the main task, and race, sex as well as age information as auxiliary tasks. When fine-tuning on the target domain $\mathcal{D}_T$, we only optimize the model parameters using the main task loss without any auxiliary tasks (same as Xie et al. (2021)), which makes other steps identical to Algorithm 1. Algorithm 2 displays the local training procedure with auxiliary tasks on one of the source clients $\mathcal{C}_{S_i}$.

---

**Algorithm 2** Domain Adaptation with Auxiliary Information (Local training on $\mathcal{C}_{S_i}$)

---

**Input**: One source domain $\mathcal{D}_{S_i}$; one source client $\mathcal{C}_{S_i}$; target model $h_T^{(r-1)}$ from the previous round $r - 1$; input images $x$, auxiliary input $z$, main task label $y$, auxiliary output label $b$; main task loss function $\ell$ and auxiliary task loss function $\ell_{aux}$; loss weight control hyper-parameter $\alpha$.

**Output**: source model $h_{S_i}^{(r)}$ at round $r$.

1: Initialize $h_{S_i}^{(r)} \leftarrow h_T^{(r-1)}$.
2: Optimize $h_{S_i}^{(r)}$ on $\mathcal{D}_{S_i}$ with $L_{aux-info}(h_{S_i}^{(r)}, x, y, z, b, \ell, \ell_{aux}, \alpha)$.
3: Send $h_{S_i}^{(r)}$ to $\mathcal{S}$

---

**Algorithm 3** Gradient Projection on $\mathcal{C}_T$

---

**Input**: $N$ source domain models $h_S = \{h_{S_i}\}_{i=1}^{N}$; target model $h_T^{(r-1)}$ from the previous round $r-1$; global model $h_{global}^{(r-1)}$ from the previous round $r-1$; target domain $\mathcal{D}_T$; target client $\mathcal{C}_T$; server $\mathcal{S}$; GP weight control variable $\beta$; number of samples of source domains $\{n_i\}_{i=1}^{N}$.

**Output**: Target model $h_T^{(r)}$ at round $r$.

1: **Step 2**: $\mathcal{S}$ received all $\{h_{S_i}^{(r)}\}_{i=1}^{N}$
2: $h_{global}^{(r)} \leftarrow h_{global}^{(r-1)}$
3: **for** $h_{S_i}$ in $\{h_{S_i}^{(r)}\}_{i=1}^{N}$ **do**
4: $\quad G_{S_i} \leftarrow h_{S_i}^{(r)} - h_{global}^{(r)}$
5: **end for**
6: Send $h_{global}^{(r)}$ and $\{G_{S_i}^{(r)}\}_{i=1}^{N}$ to $\mathcal{C}_T$
7: **Step 3**: $\mathcal{C}_T$ received $\{G_{S_i}^{(r)}\}_{i=1}^{N}$
8: $h_T^{(r)} \leftarrow h_T^{(r-1)}$
9: Optimize $h_T^{(r)}$ on $\mathcal{D}_T$ using labeled samples
10: $G_T \leftarrow h_T^{(r)} - h_T^{(r-1)}$
11: Update $h_T^{(r)}$ using Eq. 7 with $\{G_{S_i}^{(r)}\}_{i=1}^{N}$, $G_T$, $\beta$, $\{n_i\}_{i=1}^{N}$.
12: Send $h_T^{(r)}$ to $\mathcal{S}$

---

# 4 UTILIZING SOURCE DOMAIN KNOWLEDGE VIA GRADIENT PROJECTION

Leveraging auxiliary information does not consider the *importance* of each source client contributing to the target client, because we only use a simple aggregation rule. In weakly-supervised FDA, how can one better utilize the knowledge from both labeled target samples and source client models? Here, we suggest a novel *Gradient Projection* (GP) method.

**Algorithm intuition.** The small set of labeled target samples provides a useful signal on the direction of target domain optimum. Thus, during each communication round, server $\mathcal{S}$ does not aggregate the weights (Algorithm 1), but instead computes the model updates denoted by $\{G_{S_i}^{(r)}\}_{i=1}^N$ where $G_{S_i}^{(r)} \simeq h_{S_i}^{(r)} - h_{global}^{(r-1)}$ at round $r$ from all source clients $\{\mathcal{C}_{S_i}\}_{i=1}^N$. On the target client $\mathcal{C}_T$, it will perform gradient projection using *cosine similarity* on each $G_{S_i}^{(r)}$ towards the target direction $G_T^{(r)} \simeq h_T^{(r)} - h_T^{(r-1)}$, which could be computed after fine-tuning on the small set of the labeled target samples. In this way, we greedily *maximize* knowledge transfer to the target domain in each round; the projection of $G_{S_i}^{(r)}$ on $G_T^{(r)}$ could be regarded as the weight of $\mathcal{D}_{S_i}$ at round $r$. By combining $G_T^{(r)}$ with projected gradients ($\{G_{S_i}^{(r)}\}_{i=1}^N$ on $G_T^{(r)}$) controlled by a hyper-parameter $\beta$, we observe a more steady convergence towards the target direction, as noted in the experiment results outlined in Section 5.2. Figure 1b and Algorithm 3 illustrate the procedure of gradient projection.

**Details of the Gradient Projection.** We compute the cosine similarity (Eq. 4) between one source client model update $G_{S_i}^{(r)}$ and target client update $G_T^{(r)}$ for each *layer* of the model (for a finer projection). In addition, we align the magnitude of the model updates according to the number of target/source samples, batch sizes, local updates, and learning rates (more details are in Appendix A.2). To prevent negative projection, we set the threshold for function **GP** to be 0. For a certain layer $l$ of $G_{S_i}^{(r)}$ and $G_T^{(r)}$ (for simplicity, we denote them as $G_{S_i}^l$ and $G_T^l$), the cosine similarity and corresponding gradient projection result is (Eq. 5):

$$\cos(G_{S_i}^l, G_T^l) = \frac{G_{S_i}^l G_T^l}{\|G_{S_i}^l\|\|G_T^l\|} \tag{4}$$

$$\mathbf{GP}(G_{S_i}^l, G_T^l) = \begin{cases} \cos(G_{S_i}^l, G_T^l), & \text{if } \cos(G_{S_i}^l, G_T^l) > 0 \\ 0, & \text{if } \cos(G_{S_i}^l, G_T^l) \le 0 \end{cases} \tag{5}$$

The total gradient projection $P_{GP}$ from all source clients $\{G_{S_i}^{(r)}\}_{i=1}^N$ projected on the target direction $G_T$ could be computed as Eq. 6. We use $\mathcal{L}$ to denote all layers of current model updates. $n_i$ denotes the number of samples trained on source client $\mathcal{C}_{S_i}$, which is adapted from FedAvg (McMahan et al., 2017) to redeem data imbalance issue. Hence, we normalize the gradient projections according to their number of samples. Also, $+\!+_{l \in \mathcal{L}}^{\mathcal{L}}$ concatenates the projected gradients of all layers.

$$P_{GP} = +\!+_{l \in \mathcal{L}}^{\mathcal{L}} \sum_{i=0}^N \left( \mathbf{GP}(G_{S_i}^l, G_T^l) \cdot \frac{n_i}{\sum_i^N n_i} \cdot G_{S_i}^l \right) \tag{6}$$

Lastly, a hyper-parameter $\beta$ is used to incorporate target update $G_T$ into $P_{GP}$ to have a more stable performance. The final target model weight $h_T^{(r)}$ at round $r$ is thus expressed as:

$$h_T^{(r)} = h_T^{(r-1)} + (1 - \beta) \cdot P_{GP} + \beta \cdot G_T \tag{7}$$

# 5 EXPERIMENTS

We evaluate our proposed framework on three medical datasets: CheXpert (Irvin et al., 2019), MIMIC (Johnson et al., 2019), and a real-world imbalanced dataset from MIDRC tasked with COVID-19 detection from X-ray images. The data are split to represent three scenarios: (a) **CheXpert**: *balanced* data across clients and a *large* distribution shift among domains. (b) **MIMIC**:

Figure 2: Summary of the medical datasets in our experiments.

| States | Number of Samples |
|--------|-------------------|
| IL     | **16203**         |
| NC     | 3717              |
| CA     | 560               |
| IN     | 557               |
| TX     | 501               |

Table 1: Statistics of MIDRC dataset: an extremely imbalanced scenario (IL consists most samples).

*balanced* data across clients and and a *small* distribution shift among domains. (c) **MIDRC** [1]: highly *imbalanced* data across clients and a *large* distribution shift across domains. Note that this is a real-world application with data from hospitals in different locations across U.S. The first two experiments are designed to highlight the kinds of *extreme* data shift often noted in healthcare, e.g., where the same features are measured differently across hospitals (Wiens et al., 2014), or where the same variables' names refer to very different measurements/ diagnoses across health systems. Unfortunately, there are little public data illustrating these kinds of important and understudied shifts. Thus, we designed representative semi-synthetic shifts to illustrate the extent and impact of the issue.

## 5.1 EXPERIMENTAL SETUP

We provide basic information and experiment setup of three medical imaging datasets in Figure 2. More details on data splitting and implementation details are discussed at Appendix A.3.

**CheXpert** is a widely used medical imaging dataset for multi-label classification consisting of 224,316 chest radiographs from 65,240 patients (Irvin et al., 2019). We use sex, age, frontal/lateral information from patients to construct auxiliary labels and split domains by labelled condition i.e., selecting "Lung Lesion", "Edema", "Consolidation", "Pneumonia", "Atelectasis"– all of which are lung conditions, as source and target domains. Thus, the task is to predict a new lung condition based on labels of existing lung conditions.

**MIMIC** is a large dataset of 227,835 imaging studies for 65,379 patients between 2011–2016 (Johnson et al., 2019). We set domains using the race information provided and merge the result into four main categories: White, Black, Asian, Hispanic/Latino. Thus, the task is to predict conditions for a previously unobserved racial group condition based on labels collected from other groups. The distribution shifts between race domains are considered small, as we can get a high accuracy simply using FedAvg, as shown in Table 2.

**MIDRC** includes Computed Radiography (CR) images as the primary input. We evaluate the proposed framework using NC, CA, IN, TX states as source clients, and try to adapt to the target client IL which has a *large* number of unlabeled samples. The statistics of these five states are shown in Table 1. We collect race, sex, and age data as auxiliary information.

## 5.2 MAIN RESULTS

Table 2 and Figure 3 compare the target domain accuracy and convergence speed of following methods on three datasets: a) **FedAvg** only aggregates the source clients using FedAvg (McMahan et al., 2017) without fine-tuning on any target samples; b) **FedAvg_Finetune** performs fine-tuning step after source client aggregations (Algorithm 1); c) **FedAvg_Finetune_AuxInfo** includes auxiliary information during local training (Algorithm 2); d) **FedAvg_Finetune_GP** performs gradient projection during fine-tuning (Algorithm 3); e) **FedAvg_Finetune_AuxInfo_GP** combines c) and d)

---

[1]MIDRC data is semi-public, and is available by request `https://www.midrc.org/`

| | CheXpert | | MIMIC | | MIDRC | |
|---|---|---|---|---|---|---|
| | ACC | AUC | ACC | AUC | ACC | AUC |
| FedAvg | 53.28±2.44 | 54.13±3.77 | 69.95±1.57 | 76.34±3.02 | 53.56±2.33 | 52.45±4.23 |
| FedAvg_Finetune | 58.90±0.77 | 63.81±2.92 | 70.21±1.38 | 77.18±0.87 | 62.33±2.34 | 66.66±2.17 |
| FedAvg_Finetune_AuxInfo | 63.15±2.11 | 69.10±4.08 | 69.41±1.40 | 76.38±1.27 | 68.22±3.68 | 73.41±1.68 |
| FedAvg_Finetune_GP | 72.63±2.27 | 82.40±2.17 | **70.69±0.91** | **77.24±1.87** | 69.36±2.93 | 72.10±2.48 |
| FedAvg_Finetune_AuxInfo_GP | **75.61±1.17** | **83.09±2.28** | 68.68±2.16 | 75.54±2.44 | **70.28±1.68** | **74.61±3.16** |
| FADA (Peng et al., 2020) | 61.83±3.33 | 64.83±5.44 | 65.92±1.43 | 70.11±2.04 | 51.98±1.59 | 53.19±2.39 |
| KD3A (Feng et al., 2021) | 62.61±0.50 | 67.72±2.50 | 70.08±0.64 | 75.50±0.94 | 51.27±2.69 | 55.76±0.55 |
| Oracle | 75.72±1.27 | 82.51±1.15 | 71.28±0.79 | 76.29±0.94 | 84.28±1.01 | 87.62±1.54 |

Table 2: **Target domain accuracy and AUC scores** (%) on three medical datasets with comparisons with SOTA methods. Results are reported averaged across 3 trials and 95% confidence intervals.

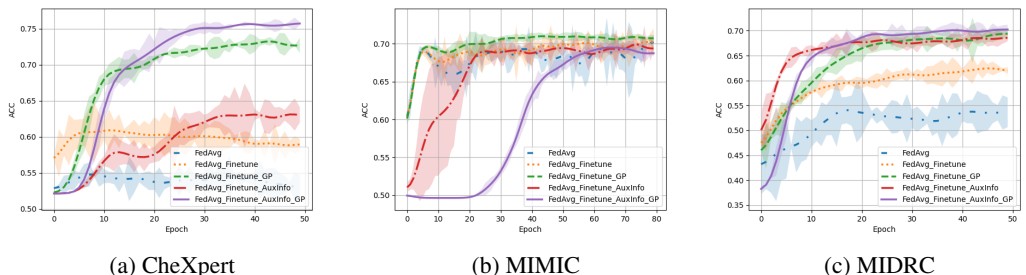

| (a) CheXpert | (b) MIMIC | (c) MIDRC |
|---|---|---|

Figure 3: **Target Domain Accuracy v.s Rounds** on (a) CheXpert (b) MIMIC (c) MIDRC datasets. For the large shift scenarios (a) and (c), AuxInfo and GP both significantly improve target domain performance. For the small shift scenario (b), GP still manages to achieve 1% boost while AuxInfo may introduce extra noise into the training procedure.

together; f) and g) **FADA** (Peng et al., 2020) and **KD3A** (Feng et al., 2021) are SOTA methods focusing on the unsupervised FDA setting; h) **Oracle** refers to a fully-supervised training on the target domain and serves as the upper bound. For all datasets, our framework generally outperforms the state-of-the-art FDA methods with a large margin and is close to the oracle performance.

**CheXpert.** For balanced and large distribution shifts, AuxInfo and GP improve around 6% and 18% of accuracy and AUC individually. Combining the two together, we can get a further $2 \sim 3\%$ gain with *close-to-oracle* performance. However, we notice that AuxInfo slows down the convergence speed of the training procedure.

**MIMIC.** For balanced and small distribution shifts, the boost compared with baseline (FedAvg) becomes small. Yet GP still manages to achieve a 1% boost with a *close-to-oracle* performance. We think when shifts are small, including auxiliary information may hinder the fine-tuning of the model, introducing extra noise during the training procedure. The variance is quite large for AuxInfo during the first several epochs, which leads to a slow convergence when combining two techniques. When the local signals are not helpful, doing GP on top enlarges the negative effect. We hypothesize that this is why it results in a slightly worsened performance.

**MIDRC.** For imbalanced and large distribution shifts, both AuxInfo and GP achieve a significant increase of 6% and 7% on the target client accuracy/AUC. $1 \sim 2\%$ extra increase is obtained when combining AuxInfo with GP. It is interesting to see when client data is imbalanced, AuxInfo actually achieves a faster convergence. We think data imbalance may require more signals coming from auxiliary information to converge. In general, data imbalance has little impact on the performance of our proposed framework. We hypothesize that is because we have normalized the client sample size when updating the model weights. In contrast, SOTA methods are not resistant to this issue.

**Computational complexity analysis.** As shown in Table 3, our proposed framework requires a small amount of additional time and space complexity compared with SOTA methods. GP has a memory cost of $O(1)$ and a time cost of $O(N \cdot m^2/l)$ (Details in Appendix A.4). The average time for calling the GP function is 0.068 seconds (pretty fast) using the ResNet-18 with $N = 4$.

| Methods | Number of parameters | Operations with extra time per epoch | Wallclock time (in seconds) |
|---|---|---|---|
| FedAvg | 5.6212E+07 | / | 30.70 |
| FedAvg_Finetune | 5.6212E+07 | Fine-tune $n_l$ labeled samples in $\mathcal{D}_{\mathcal{T}}$ | 31.53 (+0.83) |
| **Ours** | **5.6221E+07** | Fine-tune $n_l$ labeled samples in $\mathcal{D}_{\mathcal{T}}$ + optimize auxiliary tasks (Eq. 3)+ GP aggregation | **34.38 (+3.68)** |
| KD3A | 5.6212E+07 | Train all unlabeled samples in $\mathcal{D}_{\mathcal{T}}$ | 41.67 (+10.97) |
| FADA | 5.6369E+07 | Train all unlabeled samples in $\mathcal{D}_{\mathcal{T}}$ | 166.32 (+135.62) |

Table 3: **Comparison of computational efficiency**. We calculate the average time for running 1 global epoch on CheXpert with $N = 4$ source clients using the same Quadro RTX 6000 GPU. We do not consider communication and testing cost and assume the clients' training happens sequentially.

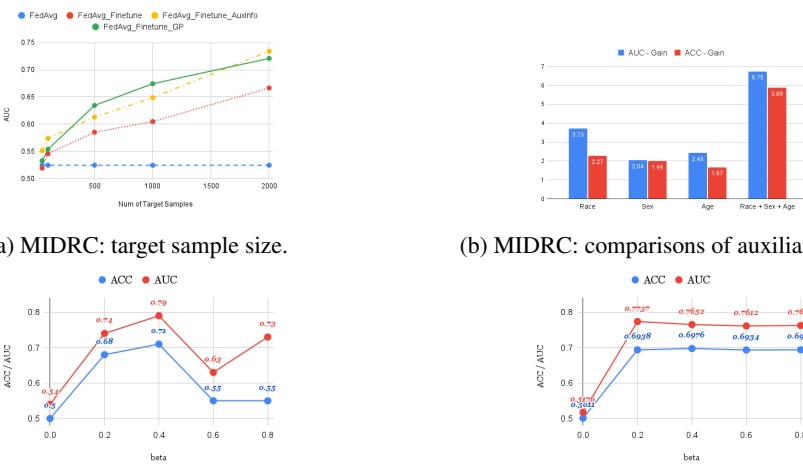

(a) MIDRC: target sample size.

(b) MIDRC: comparisons of auxiliary tasks.

(c) CheXpert: $\beta$.

(d) MIMIC: $\beta$.

Figure 4: (a) Target AUC with different sample sizes on FedAvg v.s Finetune v.s. AuxInfo v.s. GP for MIDRC. (b) Performance gain compared with FedAvg_Finetune using different auxiliary inputs. (c) and (d) display the value of $\beta$ v.s. target domain ACC/AUC for CheXpert and MIMIC.

**Effects of labeled target sample sizes.** We perform the experiment on MIDRC dataset with various target sample sizes of $(50, 100, 500, 1000, 2000)$, testing its impact on AuxInfo and GP individually, as shown in Table 4 and Figure 4a. We report AUC scores for a more accurate comparison. Generally, as we increase the number of target samples, the target domain accuracy boosts as well. Apart from that, the improvements coming from AuxInfo and GP compared with the baselines also increase when we have more labeled target samples.

|  | 50 | 100 | 500 | 1000 | 2000 |
|---|---|---|---|---|---|
| FedAvg | 52.45 | 52.45 | 52.45 | 52.45 | 52.45 |
| Finetune | 51.92 | 54.56 | 58.52 | 60.48 | 65.57 |
| AuxInfo | **55.13** | **57.38** | 61.30 | 64.88 | **72.51** |
| GP | 53.27 | 55.38 | **63.45** | **67.44** | 71.98 |

Table 4: Target domain AUC (%) on MIDRC dataset with different number of target labeled samples.

**Effects of number of auxiliary tasks.** To compare the contribution of each auxiliary task, we train FedAvg_Finetune_AuxInfo with a single auxiliary task branch of race/sex/age. In Figure 4b, we report the performance gain of each auxiliary information compared with FedAvg_Finetune. The auxiliary tasks seem to have a summed gain effect (race_gain + sex_gain + age_gain ≈ all_gain).

**Choice of controlled hyper-parameter** $\beta$ between gradient projections $P_{GP}$ and fine-tuning gradient update $G_{\mathcal{T}}$. We set $\beta = 0, 0.2, 0.4, 0.6, 0.8$ and use $50, 20$ labeled target samples on CheXpert and MIMIC datasets. We use a small number of labeled samples to better evaluate the effectiveness of GP. Figure 4c and Figure 4d present the target domain accuracies with different $\beta$ choices. When $\beta = 0$, it relies solely on the set of labeled samples to optimize the parameters. In other words, it could be regarded as not transferring any knowledge from source clients. For both large and small shift cases, We observe around $\beta = 0.4$, both the accuracy and AUC reach the highest values. Whereas, when $\beta$ is large, the performance drops severely for CheXpert but has little influence on MIMIC, for which a close-to-oracle performance is more easily attained. For the large shift case,

we find that the total gradient updates become too greedy, taking an overly-large step projecting the gradients from source clients, leading to a worsened performance, while it results in little harm for the small shift case. Therefore, we choose $\beta = 0.5$ for our experiments, though the tuning of $\beta$ may change slightly as the number of labeled target samples varies.

# 6 RELATED WORK

**Data heterogeneity and label deficiency in federated learning.** Distribution shifts between clients remains a crucial challenge in FL. Current work often focus on improving the aggregation rules: Karimireddy et al. (2020) use control variates and Xie et al. (2020b) cluster the client weights via EM algorithm to correct the drifts among clients. In medical scene, Jiang et al. (2022) and Dinsdale et al. (2022) try to mitigate local and global drifts via harmonisation. However, people usually assume the local training is fully-supervised for all clients at present. The truth is, label deficiency problem could happen in any of the clients. There recent works try to tackle label deficiency problem with self-supervision or semi-supervision for better personalized models (Jeong et al., 2020; He et al., 2021; Yang et al., 2021). Compared to them, we explore a new setting with fully-labeled source clients and one few-labeled target client, improving FDA performance under weak supervision.

**Federated domain adaptation.** There are a considerable amount of recent work on multi-source domain adaptation with unsupervised setting, with recent highlights on adversarial training (Saito et al., 2018; Zhao et al., 2018), knowledge distillation (Nguyen et al., 2021), and source-free methods (Liang et al., 2020). Peng et al. (2020); Li et al. (2020) are the first to extend MSDA into FL setting; they apply adversarial adaptation techniques to align the representations of nodes. More recently, in KD3A (Feng et al., 2021) and COPA (Wu & Gong, 2021), the server with unlabeled target samples aggregates the local models by learning the importance of each source domain, via knowledge distillation and collaborative optimization. Our work is in contrast to them that primarily focus on the unsupervised setting. Here, these methods rely heavily on both source and target data with complicated training procedure on the server. Our framework is *computationally efficient*, exploring FDA problem in a new manner with auxiliary labels available to clients.

**Auxiliary information in domain generalization.** In-N-Out (Xie et al., 2021) investigate both out-of-distribution (OOD) and in-distribution performance of using auxiliary information as inputs and outputs with self-training. In medical scene, Wang et al. (2022) find by pre-training and fine-tuning on the auxiliary tasks, one could improve the transfer performance between datasets on the primary task. They consider single source-target scenario with no labeled target data while our work focuses on federated MSDA setting with few labeled target data. Their frameworks cannot be directly adapted to our setting, since they require training on the source samples again after training on out-of-domain (target) samples. Our MLT framework properly leverages auxiliary labels in the new setting, is cheap and efficient to compute, with good improvement for large shift cases.

**Using additional gradient information in FL.** Model updates in each communication round could provide valuable insights of client convergence directions, which is mostly explored for byzantine robustness in FL. Zeno++ (Xie et al., 2020a) and FlTrust (Cao et al., 2021) leverage the additional gradient computed from a small clean training dataset on the server, which helps compute the scores of candidate gradients for detecting the malicious adversaries. In our work, we utilize the additional gradient provided by the labeled target samples for FDA problem. In a simple yet effective way, we project source gradients towards the target direction. Our results support that we transfer the knowledge from source clients to the target client with a better and more stable performance.

# 7 CONCLUSION AND FUTURE WORK

We show that including auxiliary information during local training and gradient projection during fine-tuning, can help address significant distribution shifts and label deficiency issues existing in current federated learning systems, particularly for real medical applications. Our results on healthcare datasets show our proposed framework improves FDA performance with small additional computational cost. Future work includes evaluating on the fly/ offline finetuning scheme, exploring how to select the set of labeled target samples in the real-world case to better align with the distribution of the unlabeled part, analyzing the impact of more factors related to domain discrepancy, and extending current framework to more general transfer learning setting.

## 8 REPRODUCIBILITY STATEMENT

We have provided the details of our dataset preprocessing, hyper-parameters, training scheme, and model architecture in Section 5 and in the Appendix. Also, we have uploaded the source code of our proposed framework as part of the the supplementary materials. Because access to the MIDRC data is restricted to approved users, we are unable to include the original data. We will release the code upon acceptance.

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

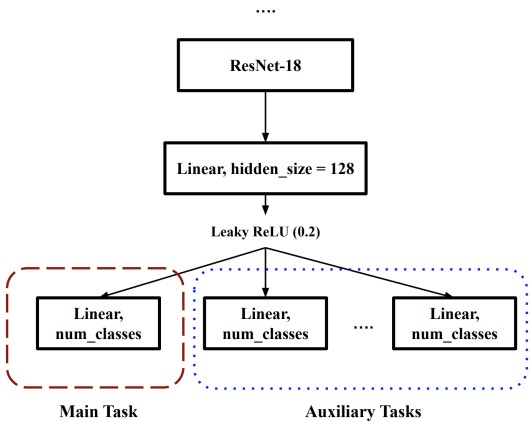

Figure 5: ResNet-18 model architecture for training auxiliary tasks.

## A APPENDIX

### A.1 NETWORK ARCHITECTURE

Figure 5 displays the ResNet-18 model architecture (He et al., 2016) used in our experiment. When training on the single classification task, the output layer of ResNet-18 only consists one branch with the main task output predictions. When training with other auxiliary tasks, we add extra branches to the model and output several classification logits, optimizing the losses of all tasks together.

### A.2 IMPLEMENTATION DETAILS OF GP

In the previous Section 4, we compute the model updates from source and target clients as $G_{S_i}^{(r)} \simeq h_{S_i}^{(r)} - h_{global}^{(r-1)}$ and $G_T^{(r)} \simeq h_T^{(r)} - h_T^{(r-1)}$, respectively. In our real training process, because we use different learning rates, training samples for source and target clients, we need to **align the magnitude of model updates**. Eq. 8 aligns the model updates from source clients to the target client and Eq. 9 combines the projection results with the target updates. We use $lr_T$ and $lr_S$ to denote the target and source learning rates; $batchsize_T$ and $batchsize_S$ are the batch sizes for target and source domains, respectively; $n_l$ is the labeled sample size on target client and $n_i$ is the sample size for source client $\mathcal{C}_{S_i}$; $r_S$ is the rounds of local updates on source clients.

$$P_{GP} = \mathbin{+\!\!+}_{l \in \mathcal{L}} \sum_{i=0}^{N} \left( \mathbf{GP}\left( \left(h_{S_i}^{(r)} - h_{global}^{(r-1)}\right)^l, \left(h_T^{(r)} - h_T^{(r-1)}\right)^l \right) \cdot \frac{n_i}{\sum_i^N n_i} \cdot \frac{\frac{n_l}{batchsize_T}}{\frac{n_i}{batchsize_S}} \cdot \frac{lr_T}{lr_S} \cdot \frac{1}{r_S} \cdot \left(h_{S_i}^{(r)} - h_{global}^{(r-1)}\right) \right) \tag{8}$$

$$h_T^{(r)} = h_T^{(r-1)} + (1 - \beta) \cdot P_{GP} + \beta \cdot (h_T^{(r)} - h_T^{(r-1)}) \tag{9}$$

### A.3 IMPLEMENTATION AND DATA SPLITTING DETAILS

**CheXpert.** We randomly sampled 4,000 source labeled samples (2,000 positive, 2,000 negative) from each domain for and 1,000 target labeled samples for fine-tuning. To create a larger distribution shift, for negative samples, we randomly sample the ones with labels "0" for that condition instead of using "No Finding" labels.

**MIMIC.** We sample 5,00 (250 with findings and 250 without findings) for each source domain and adapt to the target domain with 100 labeled samples. This small sample size is intentionally chosen to increase the difficulty of this federated domain adaptation task. For the auxiliary labels, we use sex and age information from the database.

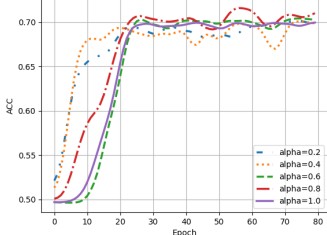

Figure 6: Choice of $\alpha$ values on including auxiliary information.

**MIDRC.** To set up a real-world case with multiple source domains, we split the CR dataset according to the zip code and select 5 states (IL, NC, CA, IN, TX) as our source and target domains. We use all labeled samples in the source domains for local training, and 2,000 target labeled samples for fine-tuning. As a real-world dataset, the number of samples are extremely imbalanced across the clients, since the dataset collects data mostly from Chicago, which potentially would introduce more distributional noise into clients.

**Setup.** We used FedAvg as the aggregation rule for baselines, and set the hyper-parameters $\alpha = 1$ and $\beta = 0.5$, source learning rate as $10^{-3}$ and target learning rate as $2 \cdot 10^{-4}$, communication rounds $r = 80$ for MIMIC dataset and $r = 50$ for MIDRC and CheXpert datasets, as well as the local training step size to be 1. We use cross-entropy losses for classification tasks and Adam optimizer (Kingma & Ba, 2014). We select around $20\% \sim 25\%$ of target domain labeled samples for fine-tuning under weak supervision. Further, we use pretrained ResNet-18 (He et al., 2016) model with last layer hidden size of 128 for the training of three datasets. When training with auxiliary tasks, we added branches to the output layer of ResNet-18 as shown in Figure 5.

### A.4 DERIVATION OF GRADIENT PROJECTION METHOD'S TIME AND SPACE COMPLEXITY

**Time complexity**: Assume the total parameter is $m$ and we have $l$ layers. To make it simpler, assume each layer has an average of $\frac{m}{l}$ parameters. Computing cosine similarity for all layers of one source client is $O((\frac{m}{l})^2 \cdot l) = O(m^2/l)$. We have $N$ source clients so the total time cost for GP is $O(N \cdot m^2/l)$.

**Space complexity**: The extra memory cost for GP (computing cosine similarity) is $O(1)$ per client for storing the current cosine similarity value.

### A.5 CHOICE OF THE LOSS WEIGHT HYPER-PARAMETER $\alpha$ FOR AUXILIARY TASKS

We observe that the convergence speed for AuxInfo is slow for small distribution shift case (MIMIC dataset). Thus, we further conduct the ablation study on MIMIC dataset for hyper-parameter $\alpha$, which controls the loss weights between the main task and auxiliary tasks. We set $\alpha = 0.2, 0.4, 0.6, 0.8$ and Figure 6 exhibits their target accuracies v.s. epochs. Setting $\alpha$ smaller may lead to a faster convergence while the final performances are almost the same for different $\alpha$ values. Thus, we set $\alpha = 1$ for all experiments for a fair comparison.

|  | Books | DVD | Electronics | Kitchen | Average |
|---|---|---|---|---|---|
| FADA | 78.10% | 82.70% | 77.40% | 77.50% | 78.90% |
| KD3A | 79.00% | 80.60% | **85.60%** | 86.90% | 83.10% |
| FedAvg | 77.93% | 80.50% | 82.19% | 84.36% | 81.24% |
| FedAvg_Finetune | 79.68% | 81.31% | 85.41% | 86.76% | 83.29% |
| FedAvg_Finetune_GP | **79.78%** | **82.35%** | 84.93% | **86.88%** | **83.48%** |

Table 5: Target domain accuracy (%) on AmazonReview dataset.

|  | Target Accuracy |
|---|---|
| FedAvg | 60.11% |
| FedAvg_Finetune | 69.82% |
| FedAvg_Finetune_GP | **79.74%** |

Table 6: Target domain accuracy (%) on Non-IID MNIST dataset.

### A.6    SUPPLEMENTARY EXPERIMENTS ON GRADIENT PROJECTION METHOD

Apart from experimenting on medical imaging datasets for GP, we also test GP on two general-purposed datasets: AmazonReview (McAuley et al., 2015) and self-generated Non-IID MNIST (Deng, 2012) datasets with imbalanced labels. We illustrate details of two datasets and present the experiment results in this section.

#### A.6.1    DATASETS

**Amazon Review.** This dataset is for a binary sentimental analysis task including four domains. By randomly choosing three of these domains as source domains and the rest one as the target domain, we train a simple CNN model for the classification task. We use 2,000 training samples (the same as KD3A (Feng et al., 2021) for a better comparison) for each source domain and 400 labeled target samples.

**Non-IID MNIST.** We adapt the Non-IID benchmark (Li et al., 2022) to construct the source and target domains for MNIST dataset in a Non-IID manner. To make the task harder, we choose the data partition with quantity-based label imbalance with only 3 classes available for each source client (though predicting for 10 classes) and we have 8 source clients in the system. For the target client, we have 10 classes with all digits with a noise-based feature imbalance, creating a shift from the source clients. We used a CNN architecture to do the experiment on classification between digits. Also, we use 100 labeled target samples and test the accuracy on 10,000 unlabeled target samples.

#### A.6.2    RESULTS

**Amazon Review.** From the result in Table 5, this dataset includes a comparatively simple task with small distribution shifts between clients, yet GP outperforms the state-of-art unsupervised KD3A by 0.38% in average target domain accuracy using 400 labeled target samples. Though our setting is different, GP has a comparable performance against the unsupervised state-of-the-art method.

**Non-IID MNIST.** The distribution shifts between clients are larger for this dataset and GP obtains a larger boost of performance compared with the previous dataset (20% for FedAvg and 10% for FedAvg_Finetune), as shown in Table 6. Hence, we can see that GP improves the target accuracy more significantly when client shifts are bigger.

