# OpenReview forum: "Weakly-Supervised Domain Adaptation in Federated Learning"
_ICLR.cc/2023/Conference — Submitted to ICLR 2023_

### Official Review · Reviewer_JHRh · 2022-10-22

**Confidence:** 5
**Correctness:** 3
**Technical Novelty And Significance:** 2
**Empirical Novelty And Significance:** 2
**Recommendation:** 5

**Clarity, Quality, Novelty And Reproducibility:**

a) Clarity: The studied problem and proposed method are not well presented and driven, in which some aspects are confusing, e.g., the update of model parameter h_T^{(r)} utilizing the scalar P_{GP} in Eq. 7.

b) Quality: The utilized technique outperforms the compared baselines.

c) Novelty: Although the studied problem weakly-supervised federated domain adaptation is somewhat novel, the used techniques leveraging auxiliary information during local training and utilizing gradient projection during fine-tuning lacks of novelty, since the former is introduced in [1] while the latter is incorporating the cosine similarity among layers for computing the aggregation weights.

d) Reproducibility: The manuscript provides sufficient experiments details, along with the codes, which boost the reproducibility.

[1] Sang Michael Xie, Ananya Kumar, Robbie Jones, Fereshte Khani, Tengyu Ma, and Percy Liang. Inn-out: Pre-training and self-training using auxiliary information for out-of-distribution robustness. In International Conference on Learning Representations, 2021.


**Details Of Ethics Concerns:**

These are no Ethics Concerns in this manuscript.

**Strength And Weaknesses:**

Pros:
1) The studied problem weakly-supervised federated domain adaptation is interesting and practical.
2) The experiments results compared with baselines confirm the effectiveness of the used techniques, i.e., utilizing the auxiliary information and gradient projection to help local training and fine-tuning, respectively.
3) The manuscript provides sufficient experiments details, along with the codes, which help the reproducibility.

Cons:
1) Although the studied problem weakly-supervised federated domain adaptation is interesting and practical, it is not well illustrated and driven in the manuscript, which can be largely improved.
2) The proposed method is not well illustrated and driven, and even some parts are confusing. For example, in Eq. (7), the update of model parameter h_T^{(r)} utilizing the scalar P_{GP} is confusing.
3) The used techniques leveraging auxiliary information during local training and utilizing gradient projection during fine-tuning lacks of novelty, since the former is introduced in [1] while the latter is incorporating the cosine similarity among layers for computing the aggregation weights.
4) The manuscript can present the visualization of the cumulative gradient projection GP for demonstrating the effectiveness of proposed method.

[1] Sang Michael Xie, Ananya Kumar, Robbie Jones, Fereshte Khani, Tengyu Ma, and Percy Liang. Inn-out: Pre-training and self-training using auxiliary information for out-of-distribution robustness. In International Conference on Learning Representations, 2021.



**Summary Of The Paper:**

The manuscript handles the weakly-supervised domain adaptation in federated learning, in which a set of source clients aiming at boosting the model performance of the target client that has few labelled samples. To achieve the problem, the manuscript proposes to utilize the auxiliary information and gradient projection to help local training and fine-tuning, respectively. The experimental results on three medical image datasets demonstrate the effectiveness of the used techniques.

**Summary Of The Review:**

Although the studied problem is interesting and practical, but the manuscript is not well presented and driven, which should be largely improved. I checked the author's rebuttals, some of them are useful and clarifying my concerns. I chose to improve my scorings.

---

> ### Author Response · Authors · 2022-11-15
> **Response to Reviewer JHRh [1/2]**
>
> We appreciate your thoughtful and valuable comments. Below we answer several specific questions.
>
> **Q: Weakly-supervised federated domain adaptation is not well illustrated and driven in the manuscript, which can be largely improved.**
>
> We have illustrated our weakly-supervised federated domain adaptation in the introduction and problem setup sections with many details. We kindly ask what specific parts you are referring to so we can further polish our work. Thank you!
>
> **Q: In Eq. (7), the update of model parameter h_T^{(r)} utilizing the scalar P_{GP} is confusing.**
>
> We are sorry for the confusion.  $P_{GP}$  is not a scalar but a gradient - we noticed that the term $G^{l}_{s_i}$ is incorrectly missing in Eq. (7). Also, for Eq. (7), we have changed it to the concatenation of gradients from all layers to reduce notation. We have fixed the equations both in the method and appendix sections, and hopefully clarified any confusion.
>
> **Q: The used techniques leveraging auxiliary information during local training and utilizing gradient projection during fine-tuning lacks of novelty, since the former is introduced in [1] while the latter is incorporating the cosine similarity among layers for computing the aggregation weights.**
>
> In our related work, we have revised and clarified the difference between our methods from In-N-Out [1] and highlighted the novelty of the GP aggregation rule. We would like to emphasize the following points:
>
> 1. For leveraging auxiliary information:
>    - Our paper setting is very different from the previous work, and one may not a-priori assume the existing methods will be effective in the new setting. They consider single source-target domain adaptation assuming target data are all unlabeled, while our work is multi-source domain adaptation in the federated setting, with few labeled target samples. To the best of our knowledge, there is no other work that leverages auxiliary information in our setting. Related works such as [1] and [2] leverage auxiliary information in a centralized setting, and both frameworks are not easily adapted to the federated setting: they require training on the source samples again after leveraging on out-of-domain (target) samples. However, the target client cannot easily access the source clients’ data in federated domain adaptation. In addition, if we pre-train on MLT frameworks locally without finetuning, auxiliary labels will not help improve the target performance compared with FedAvg. We believe the mechanics of this implementation in the federated setting are not obvious apriori.
>    - We need to consider **efficiency** constraints in the federated setting to properly leverage auxiliary labels, which is non-trivial work. Our MLT structures are cheap and quick to compute, with good improvement for large shift cases.
>
> 2. For the Gradient Projection method:
>
>    - To the best of our knowledge, there is no previous aggregation rule existing to solve the problem of our setting: aiming to achieve good performance on the target client by transferring knowledge from source clients under weak supervision. To this end, a target gradient provided by the labeled target samples is leveraged to project the useful components from source clients. Our proposed aggregation rule helps better generalize to the target client, where no source client data/models are shared, with limited additional time and space complexity (as discussed in the next question).
>
> **References:**
>
> [1] Xie, Sang Michael, et al. "In-N-Out: Pre-Training and Self-Training using Auxiliary Information for Out-of-Distribution Robustness." International Conference on Learning Representations. 2020.
>
> [2] Wang, Rongguang, et al. "Embracing the disharmony in medical imaging: A Simple and effective framework for domain adaptation." Medical Image Analysis 76 (2022): 102309.

---

> ### Author Response · Authors · 2022-11-15
> **Response to Reviewer JHRh [2/2]**
>
> **Q: The computational efficiency of the technique gradient projection.**
>
> *Space complexity:* The extra memory cost for GP (computing cosine similarity) is $O(1)$ per client for storing the current cosine similarity value.
>
> *Time complexity:* Assume the total parameter is $m$ and we have $l$ layers. To make it simpler, assume each layer has an average of $m/l$ parameters. Computing cosine similarity for all layers of one source client is $O((m/l)^2 \cdot l) = O(m^2 / l)$. We have $N$ source clients so the total time cost for GP is $O(N \cdot m^2 / l)$. The average time for calling the GP aggregation function is 0.068 seconds using the ResNet18 model with 4 source clients, which is very fast (we are doing it on GPU).
>
> **Q: The manuscript can present the visualization of the cumulative gradient projection.**
>
> For our GP method, we project the source gradients toward the target gradient direction by computing the cosine similarity layer-wise. It is hard to visualize the gradient projection results since it is a complicated set of matrices with different cosine similarities for each source client, layer, and iteration. Our experiment results already show the effectiveness of GP on various datasets.

---

> > ### Comment · Reviewer_JHRh · 2022-11-25
> > **About computational efficiency**
> >
> > Although the analysis of time complexity and running time for calling the GP aggregation function is presented, I am concerned the  difference in training and inference time between the proposed method and compared methods.

---

> > > ### Author Response · Authors · 2022-11-26
> > > **Response to computational efficiency question**
> > >
> > > Dear Reviewer,
> > >
> > > Thanks for your reply. We have provided the time and space complexity analysis compared with SOTA methods in the experiment section (Table 3 in the updated manuscript). For your reference, we copied the table here:
> > >
> > > |                                   | Model architectures                                                                          | Number of parameters | Operations with extra time per epoch                                           | Time for running 1 epoch (in seconds) |
> > > |-----------------------------------|----------------------------------------------------------------------------------------------|----------------------|--------------------------------------------------------------------------------|---------------------------------------|
> > > | FedAvg                            | ResNet18 * (N+1)                                                                                 | 5.6212E+07           | /                                                                              | 30.70                                 |
> > > | FedAvg_Finetune                   | ResNet18 *  (N+1)                                                                                  | 5.6212E+07           | Finetune on the labeled target set                                             | 31.53 (+0.83)                         |
> > > | **FedAvg_Finetune_AuxInfo_GP (Ours)** | **(ResNet18, MLT branches) *  (N+1)**                                                                 | **5.6221E+07**           | **Finetune on the labeled target set + optimize auxiliary tasks + GP aggregation** | **34.38** (**+3.68**)                        |
> > > | KD3A [2]                          | ResNet18 *  (N+1)                                                                                  | 5.6212E+07           | Train on all unlabeled target set                                              | 41.67 (+10.97)                        |
> > > | FADA [1]                          | (ResNet18, Disentangler, Domain Identifier, Reconstructor, Mutual Information Estimator) *  (N+1)  | 5.6369E+07           | Train on all unlabeled target set                                              | 166.32 (+135.62)                      |
> > > |                                   |                                                                                              |                      |
> > >
> > > We calculated the average time for running 1 global epoch for baselines and SOTA methods on the CheXpert dataset using the same Quadro RTX 6000 GPU (we are not considering communication costs, and we assume the clients’ training happens sequentially). Here we have $N=4$ source clients. The results highlight that our proposed framework requires **a small amount of additional time and space complexity**.
> > >
> > > Thanks,
> > >
> > > Authors
> > >
> > >
> > >
> > >
> > > **References:**
> > >
> > > [1] Peng, Xingchao, et al. "Federated Adversarial Domain Adaptation." International Conference on Learning Representations. 2019.
> > >
> > > [2] Feng, Haozhe, et al. "KD3A: Unsupervised Multi-Source Decentralized Domain Adaptation via Knowledge Distillation." ICML. 2021.

---

> ### Author Response · Authors · 2022-12-07
> **Follow up**
>
> Dear reviewer,
>
> We would like to thank you again for your efforts in reviewing our submission.
>
> We have tried our best to respond to your questions. If our responses address your concerns, we sincerely hope you could reconsider the scores. We will also be very happy to answer any follow-up questions. Look forward to your updates.

---

### Official Review · Reviewer_sdrU · 2022-10-23

**Confidence:** 3
**Correctness:** 3
**Technical Novelty And Significance:** 2
**Empirical Novelty And Significance:** 2
**Recommendation:** 6

**Clarity, Quality, Novelty And Reproducibility:**

The paper is well organized. Key resources (e.g., code, data) are available, and key details are well-described to reproduce the main results.  The presentation could be further improved if the contribution are more concisely and clearly presented.

**Strength And Weaknesses:**

Strengths:
（1）	The proposed framework is reasonable and well-supported by the paper content.
（2）	Experiments are comprehensive for both the the proposed module and relevant parameters.

Weaknesses:
(1)	It uses auxiliary information in multi-task learning during the local training of each source client. However, such auxiliary information has been proposed before and are widely used, where the effectiveness of such strategy is already verified.
(2)	The experiments include only ablation studies based on the FedAvg. It lacks performance comparison with similar SOTA methods.
(3)	Minor：
1)	The “labelled” is labeled in the following sentence.  “The human cost of labeling the images is expensive, thus the data are sparsely labelled.”
2)	There are two “the” in the following sentence. “Further, in addition to the the medical images.”


**Summary Of The Paper:**

The paper introduces a weakly-supervised federated domain adaptation (FDA) framework to help the problems of distribution shifts and label deficiency in federated learning systems. It utilizes auxiliary information during local training and gradient projection during fine-tuning respectively to improve performance. Experiments on medical imaging datasets and general datasets show that the proposed framework improves FDA performance.

**Summary Of The Review:**

The paper introduces two modules in weakly-supervised federated domain adaptation (FDA), but the novelty is not so big, and the experiment lacks comparison with methods of full supervision, unsupervised, or other weak supervision.

---

> ### Author Response · Authors · 2022-11-15
> **Response to Reviewer sdrU**
>
> We appreciate your thoughtful and valuable comments. Below we answer several specific questions.
>
> **Q: Auxiliary information has been proposed before and is widely used, where the effectiveness of such a strategy is already verified.**
>
> In our related work, we have revised and clarified the novelty of leveraging auxiliary information in our setting.
>
> - Our paper setting is very different from the previous work, and one may not a-priori assume the existing methods will be effective in the new setting. They consider single source-target domain adaptation assuming target data are all unlabeled, while our work is multi-source domain adaptation in the federated setting, with few labeled target samples. To the best of our knowledge, there is no other work that leverages auxiliary information in our setting. Related works such as [1] and [2] leverage auxiliary information in a centralized setting, and both frameworks are not easily adapted to the federated setting: they require training on the source samples again after leveraging on out-of-domain (target) samples. However, the target client cannot easily access the source clients’ data in federated domain adaptation. In addition, if we pre-train on MLT frameworks locally without finetuning, auxiliary labels will not help improve the target performance compared with FedAvg. We believe the mechanics of this implementation in the federated setting are not obvious apriori.
>
> - We need to consider **efficiency** constraints in the federated setting to properly leverage auxiliary labels, which is non-trivial work. Our MLT structures are cheap and quick to compute, with good improvement for large shift cases.
>
> **Q: Performance comparison with similar SOTA methods.**
>
> We have run SOTA federated domain adaptation (FDA) methods (FADA [3], KD3A [4]) which have their code public and included their performance in our updated paper. These methods focus on the unsupervised FDA setting (to the best of our knowledge, there are no baselines for our setting i.e., weakly supervised FDA). Oracle refers to a fully-supervised training on the target domain and serves as the upper bound. Our framework performs better than SOTA on all datasets and is resistant to data imbalance issues with close to oracle performance. The results are shown below:
>
> | % (ACC / AUC) | Chexpert      | MIMIC         | MIDRC         |
> | ------------- | ------------- | ------------- | ------------- |
> | FADA \[1\]    | 61.83 / 64.83 | 65.92 / 70.11 | 51.98 / 53.19 |
> | KD3A \[2\]    | 62.61 / 67.72 | 70.08 / 75.50 | 51.27 / 55.76 |
> | **Ours**          | **75.61 / 83.09** | **70.69 / 77.24** | **70.28 / 74.61** |
> | Oracle        | 75.72 / 82.51 | 71.28 / 76.29 | 84.28 / 87.62 |
>
> **Q: The contribution should be more concisely and presented.**
>
> We have revised and clarified the auxiliary information section in the related work, and also include a comparison with SOTA methods (better performance, small time and space cost) in the experiment section of the updated manuscript.
>
> **Q: Typos.**
>
> We have fixed the typos. Thanks for pointing them out!
>
> **References:**
>
> [1] Xie, Sang Michael, et al. "In-N-Out: Pre-Training and Self-Training using Auxiliary Information for Out-of-Distribution Robustness." International Conference on Learning Representations. 2020.
>
> [2] Wang, Rongguang, et al. "Embracing the disharmony in medical imaging: A Simple and effective framework for domain adaptation." Medical Image Analysis 76 (2022): 102309.
>
> [3] Peng, Xingchao, et al. "Federated Adversarial Domain Adaptation." International Conference on Learning Representations. 2019.
>
> [4] Feng, Haozhe, et al. "KD3A: Unsupervised Multi-Source Decentralized Domain Adaptation via Knowledge Distillation." ICML. 2021.

---

> > ### Comment · Reviewer_sdrU · 2022-12-09
> > **Thanks for clarifying the different task setting and novelty**
> >
> > I agree on the feedback and are happy to increase my rating.

---

> > > ### Author Response · Authors · 2022-12-09
> > > **Thank you for your positive feedback**
> > >
> > > Dear reviewer sdrU,
> > >
> > > Thanks for raising the score. It is encouraging to see that we are able to address your concerns. Thank you!

---

> ### Author Response · Authors · 2022-12-07
> **Follow up**
>
> Dear reviewer,
>
> We would like to thank you again for your efforts in reviewing our submission.
>
> We have tried our best to respond to your questions. If our responses address your concerns, we sincerely hope you could reconsider the scores. We will also be very happy to answer any follow-up questions. Look forward to your updates.

---

### Official Review · Reviewer_v8Kd · 2022-10-24

**Confidence:** 5
**Correctness:** 4
**Technical Novelty And Significance:** 4
**Empirical Novelty And Significance:** 4
**Recommendation:** 5

**Clarity, Quality, Novelty And Reproducibility:**

+ The paper is clearly written and organized.
+ The contribution is good. Using MTL and gradient projection in FDA allows leveraging secondary local tasks, and provides a different and more efficient way to aggregate/transfer knowledge from source to target.

**Details Of Ethics Concerns:**

None.

**Strength And Weaknesses:**

Strengths:
+ The new method allows leveraging inexpensive secondary tasks to improve target domain performance.
+ Gradient projection aggregation helps improve performance.
+ The empirical results show the benefit of the method, and some ablation studies are provided.
+ The supplementary material provides additional implementation details and experimental results that help support the paper.
+ The paper includes information that would make it possible to reproduce the methods and experiments (but not with MIDRC data).

Weaknesses:
 - The experimental validation is limited in some respects. There is a lack of comparison with state-of-the-art domain adaptation methods, which makes it difficult to assess the performance of the proposed method, and whether the proposed method provides an improvement.
- The gradient projection requires that all models must have the same architecture/type.
- The authors do present average results over several independent replications, using some cross-validation process.
- The proposal should be also compared with SOA methods in terms of time and memory complexity.  There should be further analysis of the impact on the performance of growing class imbalance, degree of shift, and diversity among source and target domains.
- The tables should show upper-bound results on all datasets.

**Summary Of The Paper:**

The authors propose a new method for federated domain adaptation (FDA) with application to medical imaging. to improve target domain performance, authors consider a multi-task learning (MTL) framework over the source domains. Several cheap secondary tasks are considered locally (at each source client). Such local information has been shown to improve generalization for data from the target domain. To furthermore benefit from such local signals, authors propose an aggregation of knowledge collected from source domains and transfer it to the target domain using a gradient projection.  The FDA method is validated on 3 medical datasets and compared to standard fedavg tuned and not-tuned on target domain data, and the proposed method showed good improvement.

**Summary Of The Review:**

The proposed methods can help to improve the generalization over the target domain. However, there are limited comparisons to state-of-the-art methods.

---

> ### Author Response · Authors · 2022-11-15
> **Response to Reviewer v8Kd [1/2]**
>
> We appreciate your thoughtful and valuable comments. Below we answer several specific questions.
>
> **Q: A lack of comparison with state-of-the-art domain adaptation methods and the upper bound results.**
>
> We have run SOTA federated domain adaptation (FDA) methods (FADA [1], KD3A [2]) which have their code public and included their performance in our updated paper. These methods focus on the unsupervised FDA setting (to the best of our knowledge, there are no baselines for our setting i.e., weakly supervised FDA). Oracle refers to a fully-supervised training on the target domain and serves as the upper bound. Our framework performs better than SOTA on all datasets and is resistant to data imbalance issues with *close to oracle* performance. The results are shown below:
>
> | % (ACC / AUC) | Chexpert      | MIMIC         | MIDRC         |
> | ------------- | ------------- | ------------- | ------------- |
> | FADA \[1\]    | 61.83 / 64.83 | 65.92 / 70.11 | 51.98 / 53.19 |
> | KD3A \[2\]    | 62.61 / 67.72 | 70.08 / 75.50 | 51.27 / 55.76 |
> | **Ours**          | **75.61 / 83.09** | **70.69 / 77.24** | **70.28 / 74.61** |
> | Oracle        | 75.72 / 82.51 | 71.28 / 76.29 | 84.28 / 87.62 |
>
> **Q: Comparison with SOTA methods in terms of time and memory complexity.**
>
> We calculated the average time for running 1 global epoch for baselines and SOTA methods on the CheXpert dataset using the same Quadro RTX 6000 GPU (we are not considering communication costs, and we assume the clients’ training happens sequentially). Here we have $N=4$ source clients. The extra memory cost for GP (computing cosine similarity) is $O(1)$ per client for storing the current cosine similarity value. For the time complexity of GP, assume the total parameter is $m$ and we have $l$ layers. To make it simpler, assume each layer has an average of $m/l$ parameters. Computing cosine similarity for all layers of one source client is $O((m/l)^2 \cdot l) = O(m^2 / l)$. We have $N$ source clients so the total time cost for GP is $O(N \cdot m^2 / l)$. The average time for the GP aggregation function is 0.068 seconds, which is very fast. The results below highlight that our proposed framework requires **a small amount of additional time and space complexity**.
>
> |                                   | Model architectures                                                                          | Number of parameters | Operations with extra time per epoch                                           | Time for running 1 epoch (in seconds) |
> |-----------------------------------|----------------------------------------------------------------------------------------------|----------------------|--------------------------------------------------------------------------------|---------------------------------------|
> | FedAvg                            | ResNet18 * (N+1)                                                                                 | 5.6212E+07           | /                                                                              | 30.70                                 |
> | FedAvg_Finetune                   | ResNet18 *  (N+1)                                                                                  | 5.6212E+07           | Finetune on the labeled target set                                             | 31.53 (+0.83)                         |
> | **FedAvg_Finetune_AuxInfo_GP (Ours)** | **(ResNet18, MLT branches) *  (N+1)**                                                                 | **5.6221E+07**           | **Finetune on the labeled target set + optimize auxiliary tasks + GP aggregation** | **34.38** (**+3.68**)                        |
> | KD3A [2]                          | ResNet18 *  (N+1)                                                                                  | 5.6212E+07           | Train on all unlabeled target set                                              | 41.67 (+10.97)                        |
> | FADA [1]                          | (ResNet18, Disentangler, Domain Identifier, Reconstructor, Mutual Information Estimator) *  (N+1)  | 5.6369E+07           | Train on all unlabeled target set                                              | 166.32 (+135.62)                      |
> |                                   |                                                                                              |                      |
>
> **References:**
>
> [1] Peng, Xingchao, et al. "Federated Adversarial Domain Adaptation." International Conference on Learning Representations. 2019.
>
> [2] Feng, Haozhe, et al. "KD3A: Unsupervised Multi-Source Decentralized Domain Adaptation via Knowledge Distillation." ICML. 2021.

---

> ### Author Response · Authors · 2022-11-15
> **Response to Reviewer v8Kd [2/2]**
>
> **Q: The gradient projection requires that all models must have the same architecture/type.**
>
> Indeed, this is one of the limitations of our GP method. Like most federated learning approaches (including the standard FedAvg), GP aggregates the weights, so requires the same architecture across clients. Similarly,  finetuning of a pre-trained model in domain adaptation usually uses the same model architectures. In our future work, we hope to handle the more challenging scenario when model architectures are different locally (sometimes known as general transfer learning).

---

> ### Author Response · Authors · 2022-12-07
> **Follow up**
>
> Dear reviewer,
>
> We would like to thank you again for your efforts in reviewing our submission.
>
> We have tried our best to respond to your questions. If our responses address your concerns, we sincerely hope you could reconsider the scores. We will also be very happy to answer any follow-up questions. Look forward to your updates.

---

> ### Author Response · Authors · 2022-12-11
> **Discussion stage is about to end**
>
> Dear Reviewer v8Kd,
>
> Thank you again for your constructive and insightful feedback. We have provided performance and computational efficiency analysis compared with SOTA methods in our updated draft. We sincerely hope our rebuttal can address your questions and concerns. Since the discussion stage is about to end, we really hope to hear from you. We are also very willing to answer any follow-up questions. Thanks a lot!
>
> Best, Authors

---

### Official Review · Reviewer_1NEn · 2022-11-05

**Confidence:** 4
**Correctness:** 3
**Technical Novelty And Significance:** 2
**Empirical Novelty And Significance:** 2
**Recommendation:** 5

**Clarity, Quality, Novelty And Reproducibility:**

The manuscript is clear to read. The novelty is marginal. It is not clear if the code will the shared with the public, but it seems it could be reproducible.

**Strength And Weaknesses:**

Strength:
- The manuscript focuses on an important setting and introduces two main methods to solve the problem. The third experiment shows good results in a real-work case.

Weakness:

- The first two experimental settings are less realistic, especially the first one, the domain differences are based on the lung conditions. The second experiment's results are incremental. It is also not clear why the proposed method has a much slower convergence on MIMIC in figure 3. The last experimental setting is more realistic, with locations as different domains. I think the first two experiments are not beneficial to support the conclusion.
- The related work could be better discussed. How to aggregate local gradient information into a global model is well-studied and should be discussed in more detail.

**Summary Of The Paper:**

The manuscript focuses on the domain adaptation issue in a federated learning setting, where the target client has limited labeled data, whereas the source data are more available with auxiliary information. The major contribution includes leveraging auxiliary information on source clients and also a gradient projection to aggregate the gradient direction from multiple source clients. The experiments were conducted on three chest X-ray datasets.

**Summary Of The Review:**

The contribution of the paper is limited in both novelty and empirical contributions. I recommend revising the paper and resubmitting it to other conferences.

---

> ### Author Response · Authors · 2022-11-15
> **Response to Reviewer 1NEn [1/2]**
>
> We appreciate your thoughtful and valuable comments. Below we answer several specific questions.
>
> **Q: The first two experimental settings are less realistic, especially the first one, the domain differences are based on the lung conditions.**
>
> We emphasize that the first two experiments are designed to highlight the kinds of “extreme” data shift often noted in healthcare, e.g., where the same features are measured in different ways across hospitals [1], or where the same variables' names refer to very different measurements or diagnoses across health systems. This occurs because, unlike many other industries, healthcare in the US is highly heterogeneous (e.g., HCA, the largest consortium of hospitals covers <2% of the market [2,3]), thus many variables are not standardized [4, 5]. Unfortunately, there are little public data illustrating these kinds of important and understudied shifts, so (in collaboration with domain experts), we designed representative semi-synthetic shifts to illustrate the extent and impact of the issue. We have clarified this framing in the updated version.
>
> **Q: The second experiment's results are incremental. It is also not clear why the proposed method has a much slower convergence on MIMIC in figure 3.**
>
> Though the improvement is incremental for the MIMIC dataset with small shifts, our method already achieves a performance close to oracle (fully-supervised training on the target data). We included the MIMIC results for completeness, and to illustrate important limitations of our approach. We believe this is essential for credible scientific advancement in this area. When the shifts are small, as in the MIMIC case, leveraging auxiliary labels may introduce noise instead of helping generalization. The variance is quite large (as shown in figure 3) for MIMIC using auxiliary information during the first several epochs, which leads to a slow convergence when combining two techniques. When the local signals are not helpful, doing GP on top enlarges the negative effect. We hypothesize that this is why it results in a slightly worsened performance. We have clarified this point in our updated paper.
>
> **Q: The first two experiments are not beneficial to support the conclusion.**
>
> As noted, the first two experiments are designed to highlight and mitigate an important and often overlooked practical issue. We show our framework boosts performance greatly when client shifts are large and is resistant to data imbalance issues. Our Appendix includes additional experiments on general-purpose datasets without auxiliary labels, which further support the conclusion. To further strengthen the main claims, we have added experiments that compare our approach to oracle (fully-supervised training on the target data)  and additional SOTA federated domain adaptation (FDA) methods, which focus on the unsupervised FDA setting (to the best of our knowledge, there are no baselines for our setting i.e., weakly supervised FDA). Our framework performs better than SOTA on all datasets and is resistant to data imbalance issues with close to oracle performance.
>
> | % (ACC / AUC) | Chexpert      | MIMIC         | MIDRC         |
> | ------------- | ------------- | ------------- | ------------- |
> | FADA \[6\]    | 61.83 / 64.83 | 65.92 / 70.11 | 51.98 / 53.19 |
> | KD3A \[7\]    | 62.61 / 67.72 | 70.08 / 75.50 | 51.27 / 55.76 |
> | **Ours**          | **75.61 / 83.09** | **70.69 / 77.24** | **70.28 / 74.61** |
> | Oracle        | 75.72 / 82.51 | 71.28 / 76.29 | 84.28 / 87.62 |
>
> **References:**
>
> [1] Wiens, Jenna, John Guttag, and Eric Horvitz. "A study in transfer learning: leveraging data from multiple hospitals to enhance hospital-specific predictions." Journal of the American Medical Informatics Association 21.4 (2014): 699-706.
>
> [2] https://www.statista.com/statistics/245010/top-us-for-profit-hospital-operators-based-on-number-of-hospitals/
>
> [3] https://en.wikipedia.org/wiki/Hospital_network
>
> [4] Adnan, Kiran, et al. "Role and challenges of unstructured big data in healthcare." Data Management, Analytics and Innovation (2020): 301-323.
>
> [5] Osarogiagbon, Raymond U et al. “Institutional-Level Differences in Quality and Outcomes of Lung Cancer Resections in the United States.” Chest vol. 159,4 (2021): 1630-1641. doi:10.1016/j.chest.2020.10.075.
>
> [6] Peng, Xingchao, et al. "Federated Adversarial Domain Adaptation." International Conference on Learning Representations. 2019.
>
> [7] Feng, Haozhe, et al. "KD3A: Unsupervised Multi-Source Decentralized Domain Adaptation via Knowledge Distillation." ICML. 2021.

---

> ### Author Response · Authors · 2022-11-15
> **Response to Reviewer 1NEn [2/2]**
>
> **Q: The related work could be better discussed. How to aggregate local gradient information into a global model is well-studied and should be discussed in more detail.**
>
> We have updated the gradient information section in our related work to better illustrate GP’s novelty. There are indeed many gradient aggregation rules in the literature, but our setting is different from the normal FL setting. To the best of our knowledge, there are no previous aggregation rules to solve the problem of our setting: aiming to achieve good performance on the target client by transferring knowledge from source clients under weak supervision. To this end, a target gradient provided by the labeled target samples is leveraged to project the useful components from source clients. Our proposed aggregation rule helps better generalize to the target client, where no source client data/models are shared, with limited additional time and space complexity. We request the reviewer include additional references in case we have missed any.
>
> **Q: It is not clear if the code will the shared with the public.**
>
> The code is already included and shared in the supplementary material. We plan to clean it up and release it upon acceptance. We have noted this in the updated version.

---

> ### Author Response · Authors · 2022-12-07
> **Follow up**
>
> Dear reviewer,
>
> We would like to thank you again for your efforts in reviewing our submission.
>
> We have tried our best to respond to your questions. If our responses address your concerns, we sincerely hope you could reconsider the scores. We will also be very happy to answer any follow-up questions. Look forward to your updates.

---

> ### Author Response · Authors · 2022-12-11
> **Discussion stage is about to end**
>
> Dear Reviewer 1NEn,
>
> Thank you again for your constructive and insightful feedback. We have 1. clarified and highlighted the novelty of the GP aggregation rule and 2. clarified the healthcare heterogeneity reason which motivates experiments on CheXpert and MIMIC datasets in our updated draft. We sincerely hope our rebuttal can address your questions and concerns. Since the discussion stage is about to end, we really hope to hear from you. We are also very willing to answer any follow-up questions. Thanks a lot!
>
> Best, Authors

---

### Author Response · Authors · 2022-11-15
**General Response**

We sincerely thank all reviewers for their insightful and helpful comments, so we can greatly improve the paper. We have updated our draft to address the concerns and suggestions from reviewers, including having more experiment results and clarifying/improving the writing. Our major changes are summarized as follows:

**1. Provided results on SOTA methods FADA [1], KD3A [2], and upper bounds.** (reviewers v8Kd and sdrU)

Our proposed framework achieves a *close-to-oracle* performance (better than SOTA methods) and is resistant to data imbalances.

**2. Added time and space complexity analysis compared with SOTA methods and the GP aggregation rule in the experiment section.** (reviewers v8Kd and JHRh)

Our framework requires limited additional space and time complexity, compared with SOTA methods. The GP function runs for ~0.06 seconds per call for ResNet18 architecture and 4 source clients with $O(1)$ extra space complexity.

**3. Clarified and highlighted the novelty of leveraging auxiliary information and the GP aggregation rule in the related work section.** (reviewers 1NEn, sdrU, and JHRh)

**4. Clarified the healthcare heterogeneity reason which motivates experiments on CheXpert and MIMIC datasets.** (reviewer 1NEn)

**5. Modified notations and typos when this led to misunderstandings.**

**6. Changed title to “Weakly-supervised Domain Adaptation in Federated Learning for Healthcare”.**

We hope our revisions address the concerns and clarify any misunderstandings. We would be happy to make future changes or reply to more questions if there are any.

**References:**

[1] Peng, Xingchao, et al. "Federated Adversarial Domain Adaptation." International Conference on Learning Representations. 2019.

[2] Feng, Haozhe, et al. "KD3A: Unsupervised Multi-Source Decentralized Domain Adaptation via Knowledge Distillation." ICML. 2021.

---

### Decision · Program_Chairs · 2023-01-20

**Decision:**

Reject

**Justification For Why Not Higher Score:**

Thanks for the detailed review and revision of the manuscript. The authors' feedback clarified some of the concerns raised by the reviewers in their initial reviews. Nevertheless, the overall contribution is rather minor and novelty is limited. Therefore, I cannot recommend the acceptance of this work.




**Justification For Why Not Lower Score:**

N/A

**Metareview: Summary, Strengths And Weaknesses:**

Summary:
This paper discusses domain adaptation in the federated learning setup under weak supervision. The key idea is to leverage auxiliary information on source clients and aggregate gradients from multiple source clients.

Strength:
The problem setting is novel and interesting. The experiment on computed radiography images is a nice challenge.

Weakness:
Novelty is limited and overall contributions are marginal.